# Glycolipid recognition and binding by Siglec-6 hinges on interactions with the cell membrane
Silvia D'Andrea[1], Edward N. Schmidt[2], Duong Bui [ID][2], Ojas Singh [ID][1], Ling Han[2], Lara K. Mahal [ID][2], John S. Klassen [ID][2], Matthew S. Macauley [ID][2] & Elisa Fadda [ID][3] ✉

Sialic acid-binding immunoglobulin-type lectins (Siglecs) regulate immune response through interactions with sialylated glycans on glycoproteins and glycolipids. Human Siglecs count 14 unique proteins and in all of those the recognition and binding of the sialic acid on the glycan target involves a conserved, or canonical, Arg residue. For a subset of human Siglecs, namely MAG, Siglec-6, and Siglec-11, this Arg appears not to be essential, suggesting that a different binding mechanism may be at play. In this work, we used all-atom molecular dynamics (MD) simulations, binding assays, and mutagenesis to investigate the structural, mechanistic and energetic details of the binding of Siglec-6 to monosialylated gangliosides. Our results show that Siglec-6 relies only partially on its conserved Arg122 for recognition of membrane-bound gangliosides and that it supplements its binding free energy through interactions with the phospholipids in the membrane surrounding the target epitope. We confirmed by mutagenesis assays that the loss of the key residues (Lys 126 and Trp 127) for membrane interaction abrogates binding. These results provide a step-change in our understanding of the diversification of human Siglecs as molecular precision tools to bind specific sialosides by adapting their structure to the biological environment where these are found.

Sialic acid-binding immunoglobulin (Ig)-like lectins (Siglecs) are immunoregulatory transmembrane receptors that recognise sialylated glycoconjugates. Siglecs modulate immune responses by triggering inhibitory or activating signals, maintaining immune homoeostasis[1–4]. The ability to distinguish self from non-self through sialic acid recognition makes Siglecs critical regulators of immune function and attractive therapeutic targets for autoimmune diseases, cancer, and infections[5], as well as for the development of chimeric antigen receptor (CAR) T-cell immunotherapeutic strategies[6–8]. Humans Siglecs count 14 distinct members that are commonly classified into two groups, see Fig. 1. Evolutionarily conserved Siglecs[9], here labelled as Group 1, are present in all mammalian species[10] and have a low degree of homology, i.e. 25–30% sequence identity, namely sialoadhesin (Siglec-1), CD22 (Siglec-2), myelin-associated glycoprotein (MAG or Siglec-4), and Siglec-15. Group 2 comprises CD33-related Siglecs (CD33rSiglecs), which are not conserved across species, but share a higher sequence similarity (~50–99%)[1], suggesting that diversification could have occurred in response to species-specific selective pressures[10].

In terms of 3D structure, human Siglecs are characterised by a different number of Ig C2-set domains terminating with an Ig V-set domain that contains the canonical binding site, where a conserved Arg binds sialosides[1], see Fig. 1. The V-set domain includes a structurally flexible C-C' loop, characterised by a high sequence variability across Siglecs. Both of these features allow the C-C' loop to play a critical role in modulating recognition[1] and contribute to the Siglecs' binding specificity[5,11].

Target sialosides can be presented to Siglecs within a wide variety of different structural contexts, namely functionalising glycans linked to membrane bound or soluble glycoproteins, terminating the chains of membrane-embedded glycolipids or on secreted oligosaccharides[12,13]. In most Siglecs, the loss of the conserved Arg in the V-set domain results in loss of binding, except in Siglec-6, where binding is only weakened by the loss of the canonical Arg, as demonstrated in recent work by some of us[14]. Earlier work indicates that MAG and Siglec-11[15–18] can also operate independently of the canonical Arg, suggesting an evolution of alternative binding mechanisms in selected members of the Siglec family. To this end, previous work[14] by some of us suggested the involvement of a non-canonical Arg in the V-set domain in the binding mechanism of Siglec-6, spearheading further investigation.

[1]Department of Chemistry, Maynooth University, Maynooth, Ireland. [2]Department of Chemistry, University of Alberta, Edmonton, AB, Canada. [3]School of Biological Sciences, University of Southampton, Southampton, UK. ✉e-mail: elisa.fadda@soton.ac.uk

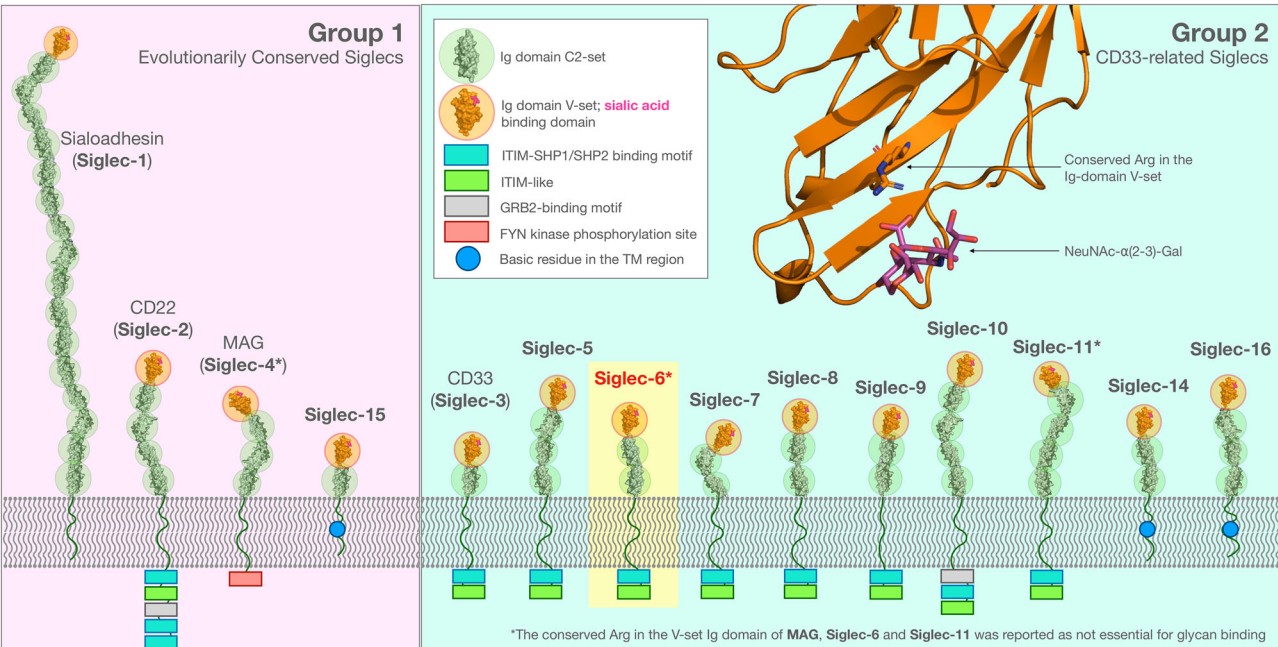

**Fig. 1 | Human Siglecs have remarkably similar architectures.** Schematic representation of human Siglecs. Evolutionarily conserved Siglecs (Group 1) are shown on the left-hand side panel (pink box). The CD33-related Siglecs (Group 2) are shown on the right-hand side panel, (light blue box). The Siglecs extracellular domains are represented by a composition of Ig domains (C2-set) rendered as a solvent accessible surface, highlighted within green circles. The terminal Ig V-set binding domain is highlighted within an orange circle with a bound sialic acid in magenta (PDB 1OD9). Cytoplasmic domains are indicated with boxes and labelled according to the legend. Siglec-6 is highlighted within a yellow box as it is the focus of this work. All structural elements in this image were rendered with pymol (www.pymol.org) and incorporated in an original design based on Fig. 1 in ref. 2. In the insert on the top right-hand side, the conserved Arg is shown in the 3D structure of the Ig domain V-set of Siglec-6 (orange cartoons) bound to the NeuNAc-a(2-3)-Gal fragment (sticks with purple C atoms, red O atoms and blue N atoms) as an example.

In this work, we used a combination of computational and molecular biology approaches to investigate the recognition and binding specificity of Siglec-6 for monosialylated gangliosides, namely GM1, GM2 and GM3, which are known Siglecs ligands[12,14]. Molecular dynamics (MD) simulations have been used extensively and successfully to investigate at the atomistic and molecular levels of details the distribution and dynamics of gangliosides, and other glycolipids, in lipid bilayers[19–21]. Coarse graining (CG) methods have been particularly useful to explore the enormous complexity of such highly dynamic systems at biologically relevant timescales, where membranes can bear a wide range of chemically diverse lipids, in different concentrations, with different structure flexibilities and diffusion properties[22]. Although the CG force fields sophistication has been greatly improved recently[23], allowing users to obtain important insights into membrane composition, structure and biology[24,25], in this work we needed an atomistic approach for our simulation to characterise the Siglec-6 recognition mechanism and how this facilitates a discrimination between structurally and chemically similar monosialylated gangliosides. The membrane used in the simulations is set to match the experimental composition, selected based on the results of extensive testing in earlier work[14] with varied lipid compositions and concentration of cholesterol.

We first gathered a detailed understanding of the binding mechanism of Siglec-6 at the atomistic-level of detail through extensive MD simulations. Based on structural analysis and on the results of MD simulations we were able to exclude the role of any other Arg residue in the V-set domain of Siglec-6. Instead, we found that the interaction between the Siglec-6 and the gangliosides still involves the canonical binding site, with the conserved Arg122 coordinating the carboxylic group of the sialic acid, but not continuously. This occurs because the interaction with Arg122 is structurally stabilised and energetically supplemented by an additional contact that Siglec-6 establishes with the lipid bilayer through residues Trp127 and Lys126, see Fig. 2a. Mutagenesis and cell binding assays, both on liposomes and on nanodisks, support the atomistic-scale insight, also demonstrating a unique interaction between Siglec-6 and the bilayer devoid of GM1. Additionally, we show through Concentration Independent (COIN)-Catch-and-Release (CaR)-native mass spectrometry (nMS) assays[26] that the binding of free (not membrane-linked) oligosaccharides hinges entirely on the canonical Arg122. In the following sections we present the results and discuss the biological implications of the unique specialisation of Siglec-6 in the recognition and binding of membrane-bound glycan epitopes with a direct comparison to other members of the Siglecs family.

## Results

### Orientation and accessibility of the ganglioside epitopes in the bilayer

We used all-atom classical MD simulations to investigate the mechanism that regulates ganglioside recognition and binding in Siglec-6 and to determine its ligand preference. Earlier work by some of us[14] shows that while Siglec-6 binds GM1 in an Arg-independent manner, it does not bind GM2 or GM3, which differ from GM1 by the lack of the terminal Gal and Gal-β(1-3)-GalNAc, respectively, see Fig. 2b. To verify that this effect was not determined by a different degree of exposure of the sialic acid through the membrane in the different gangliosides, we ran 1 μs MD simulations of isolated GM1 and GM3, as the two extremes in the monosialylated ganglioside series, in duplicates, with one ganglioside embedded in the upper and one in the lower leaflet of the bilayer, see Fig. 2d. The analysis of the tilt angle (θ) describing the orientation of the Neu5Ac along the trajectory relative to an axis perpendicular to the plane of the bilayer, see Fig. 2c, shows that the Neu5Ac is equally exposed in GM1 and GM3, and thus equally accessible for recognition and binding. These results are in excellent agreement with previous work[27].

### Structure of the Siglec-6/GM1 complex and binding mechanism

From a representative conformer of the GM1 embedded in the membrane obtained from the MD simulations described above, we built the 3D

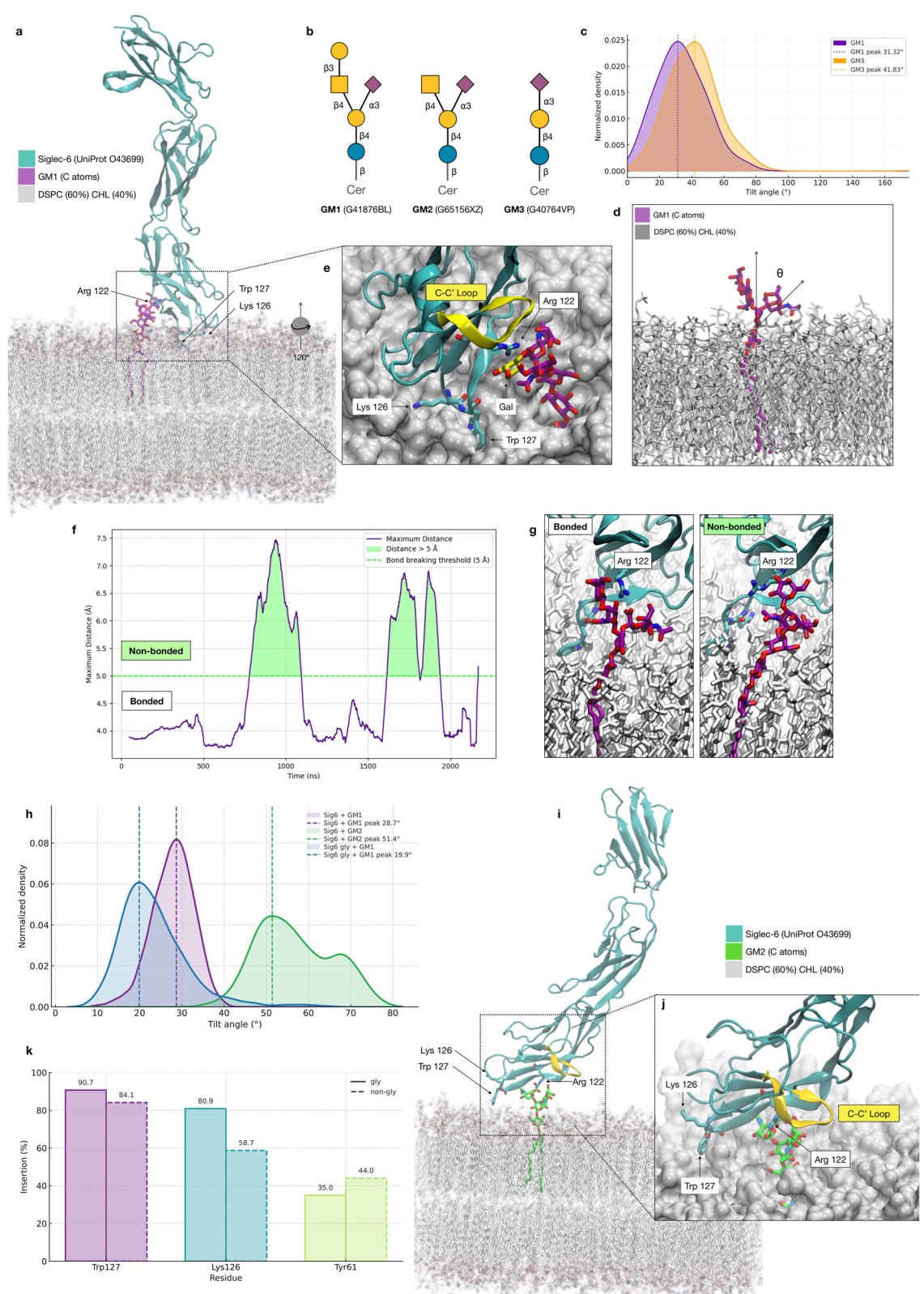

structure of the complex with Siglec-6, with the canonical Arg122 involved in a salt bridge with the Neu5Ac of the ligand. To date, no experimental 3D structure of Siglec-6 is available in public repositories, however, the common architecture and high degree of sequence and structure homology across Siglecs support confident structure predictions by machine learning (ML) leveraging on existing structural data in the RCSB PDB. In this work we used the AlphaFold (AF)[28] model AF-O43699-F1 of Siglec-6 deposited in the EBI-EMBL AF protein structure database[29], with 69% of the residues predicted with very high confidence (pLDDT > 90) and 26% with high confidence (pLDDT > 70), see Fig. S1. Structural alignment of the AF model with the high-resolution crystal structure of Siglec-3 (PDB 7AW6 at 1.95 Å), which shares 60% sequence identity with Siglec-6, results in an

**Fig. 2 | Trp 127 and Lys 126 in the V-set domain of Siglec-6 are crucial for recognition and binding of GM1, but not of GM2 or GM3, when embedded in a lipid bilayer. a** 3D structure of Siglec-6 in complex with GM1 embedded in a lipid bilayer with composition distearoylphosphatidylcholine (DSPC) 60% and cholesterol 40%. In this representative snapshot from the MD trajectory, collected at 0.715 µs from the equilibrated MD ensemble, Arg 122 is engaged in a salt bridge with the Neu5Ac of GM1. The protein is represented with cyan cartoon rendering, the GM1 with sticks and C atoms in purple, O in red and N in blue. DSPC and cholesterol are rendered with semi-transparent sticks, with C atoms in grey, O in red, N in blue and P in yellow. Key residues are labelled with numbering corresponding to the human Siglec-6 (UniProtID O43699). **b** Structures of GM1, GM2 and GM3 represented with the SNFG nomenclature. Labels below each structure include the oligo-saccharides GlyTouCan IDs. **c** Kernel Density Estimates (KDE) analysis of the tilt angle values measured though the 1.0 µs MD trajectories ran for isolated GM1 (purple) and GM3 (orange). KDE maxima are 31.32° and 41.83° measured for GM1 and GM3, respectively. **d** 3D structure of an isolated GM1 molecule (sticks with C atoms in purple, O in red and N in blue) embedded in one of the bilayer leaflets (sticks with all atoms in grey) used to represent the axes used to measure the tilt angle (θ) indicating the orientation relative to the bilayer of the Neu5Ac and thus its accessibility. **e** Close-up view on the Siglec-6 binding site obtained through a counterclockwise rotation of approximately 120° relative to the structure in **a**. Key residues are labelled, while the embedding of the Trp127 sidechain in the bilayer is highlighted by a surface rendering of the lipids. The terminal Gal of GM1 is highlighted with C atoms in yellow for his role in orienting the C-C' loop in the bound complex, also shown in yellow. **f** Time evolution along the MD trajectory of the distance (Å) between the Arg122 and the Neu5Ac carboxylate group. Data points correspond to the largest distance value calculated between four pairs of atoms R122-

NH1(2) and O11(12)-Neu5Ac. A distance of 5 Å was chosen as a threshold for the formation of the salt bridge. **g** Comparison of the canonical binding site of Siglec-6 in complex with GM1, the image on the left shows a snapshot from the MD simulation when Arg122 is engaged in the salt bridge, collected at 0.715 µs, while the image on the right shows a snapshot when the salt bridge is broken, collected at 1.028 µs. **h** KDE analysis of the tilt angle (°) indicating the orientation of the Ig V-set domain relative to the membrane plane, in purple and in green for the complexes with the non-glycosylated Siglec-6 and GM1 and GM1, respectively, and in blue for the complex with the glycosylated Siglec-6 and GM1. The normal line passes through the Ig V-set centre of mass (COM), while the Ig V-set axis is set to pass between the COM and an auxiliary point located at 10 Å from the COM obtained from the principal component 1 from the dynamics of the Cα atoms, representative of the spread along the eigenvector's direction. **i** 3D structure of Siglec-6 in complex with GM2 (sticks with C atoms in green) embedded in a lipid bilayer with composition distearoylphosphatidylcholine (DSPC) 60% and cholesterol 40%. In this representative snapshot from the MD trajectory, collected at 1.0 µs from the equilibrated MD ensemble, Arg 122 is still engaged in a salt bridge with the Neu5Ac of GM2, but the Siglec-6 is not embedded in the bilayer through the Trp 127 and Lys 126 due to a conformational change of the C-C' loop (yellow cartoon section). Colouring described in the legend and rendering style as in panel a). **k** Insertion (%) of the sidechain COM of the Trp127, Lys126 and Tyr61 below the membrane surface calculated along the MD trajectories for the complex with GM1 with the non-glycosylated Siglec-6 (full line) and the glycosylated Siglec-6 (dashed line). **j** Close-up view on the Siglec-6 binding site with key residues labelled. Molecular rendering done with Visual Molecular Dynamics[55] (VMD; https://www.ks.uiuc.edu/Research/vmd/). Tilt angle analysis and other graphs done with *matplotlib* (https://matplotlib.org/) and *seaborn* libraries in python (https://seaborn.pydata.org/).

RMSD value of 0.704 Å calculated over backbone atoms. The Siglec-6/GM1 complex was built by structural alignment of the membrane bound GM1 from the MD simulations to the Neu5Ac in the sialoside analogue bound to Siglec-3 in the crystal structure (PDB 7AW6), to which the Siglec-6 was aligned. As an important note, the orientation of the Siglec-6 was selected to complement the chosen equilibrium conformation of the GM1 ligand embedded in the membrane and no alteration to that were made to enhance structure complementarity. Comparison between the resulting Siglec-6/GM1 complex and the Siglec-3/sialoside-analogue complex shows that the glycan in the sialoside analogue is rotated by approximately 180° relative to the membrane bound GM1. This suggests that free and bound sialosides could potentially bind Siglecs in different orientations, due to the C2V symmetry of the carboxylate group and the broad accessibility of the Siglecs canonical binding site, as discussed further in the following sections.

The Siglec-6 sequence (453 aa) carries seven *N*-glycosylation sequons, with only one in the V-set domain at N103. Although we have no information on the glycosylation of Siglec-6 from mast cells or memory B-cells, earlier work[30] indicates that occupancy of these sites is low or partial in normal (non-aberrant) syncytiotrophoblasts, while it appears to be higher in recombinant constructs (317 aa). Molecular weights of 50 kDa are reported for Siglec-6 from normal pregnancy and preeclampsia placenta cells lysates[30], which corresponds to a low degree of *N*-glycosylation. In pre-eclampsia placental lysates, molecular weights of up to 70 kDa are found, which suggests *N*-glycosylation at potentially all seven sites of Siglec-6. Molecular weights from 57 up to 90 kDa are reported on commercial sites for recombinant products. In the absence of more detailed insight on the nature and occupancy of the sites, we ran simulations on two 3D models of the Siglec-6/GM1 complex, one with a non-glycosylated Siglec-6 and the other with a fully glycosylated Siglec-6 as a control, with all sites occupied with a biantennary mono-galactosylated complex *N*-glycan (GlyTouCan ID G99129GB; 1478.5 Da). Below we focus our attention on the results obtained for the non-glycosylated system, as it corresponds more closely to Siglec-6 expression in normal placental syncytiotrophoblasts. The details on the structure and dynamics of the complex with the fully *N*-glycosylated Siglec-6 are included in the Supplementary Material. A comparison between the results obtained show that *N*-glycosylation of Siglec-6 does not affect binding selectivity nor the mechanism.

After a multi-stage structure equilibration, see Methods section in the Supplementary Material for details, the conformational dynamics of the Siglec-6/GM1complex was analysed with MD simulations ran with a deterministic sampling scheme. We ran a single trajectory of 2.4 µs with the non-glycosylated Siglec-6, and two uncorrelated replicas of for 1.1 µs and 1.0 µs with the *N*-glycosylated Siglec-6 to explore the stability of the interactions and to monitor functionally relevant conformational changes, energetically accessible at room temperature. The results obtained indicate that, (1) the Neu5Ac of GM1 forms a salt bridge with the canonical Arg122, and (2) this interaction is structurally and energetically supplemented by the insertion in the membrane of the Siglec-6 V-set domain through the Trp127 indole sidechain and through ancillary binding of the phospholipid heads by the adjacent Lys126, see Fig. 2a. Furthermore, the terminal Gal of GM1 interacts with the C-C' loop (aa 70 to 75) through hydrogen bonds with Asp70 and Glu73. In the context of this interaction network, the salt bridge between the canonical Arg122 and the Neu5Ac of GM1 appears to be energetically dispensable at room temperature. Indeed, along the MD trajectory we observe that Arg122 interrupts its contact with Neu5Ac for significant amounts of sampling time, see Fig. 2f and g Population analysis shows that the salt bridge is occupied only 68.5% of the total simulation time. After these intervals, the salt bridge gets restored with no disruption of the Siglec-6 bound conformation, as shown by orientation of the Ig V-set domain relative to the membrane, see Fig. 2h, which is primarily stabilised by its interactions with the membrane, supported by contacts between the C-C' loop and the terminal Gal of GM1. The analysis of the tilt angle in Fig. 2h shows that the orientation of the Ig V-set in the complex with the fully glycosylated Siglec-6 is slightly more orthogonal relative to the non-glycosylated, except when contact with GM1 is lost, indicated by the tail in the KDE distribution. The results of the analysis of the degree of insertion of the Ig V-set residues in the membrane is shown in the bar chart in Fig. 2k, where we measured the (%) of the frames in which the centre of mass of the sidechain was below the membrane surface. The results clearly indicate the contributions of Lys126 and Tyr127, in both glycosylated and non-glycosylated Siglec-6 complexes. A minor contribution to binding could be given by Tyr61, which centre of mass is only sporadically found below the membrane surface when the Ig V-set is in a bound conformation. Yet its sidechain inserts into the bilayer to a larger extent when the Siglec-6 detaches from the ligand see Fig. S2 for the timeframe trace.

## The structure of the ganglioside determines Siglec-6 binding selectivity

Earlier work by some of us[14] shows that Siglec-6 binds GM1 selectively when embedded in a membrane, but not GM2 or GM3. Our results show that the terminal Gal of GM1 likely contributes to the stability of the complex through interactions with the C-C' loop. To test if our binding model supports the earlier experimental data and if the interaction between the C-C' loop and the terminal Gal is indeed critical, we built the 3D structure of the Siglec-6/GM2 complex from a conformation selected from the equilibrated ensemble of the Siglec-6/GM1 complex, by removing the terminal Gal of GM1, and thus biasing the new model system towards a stable conformation. We ran two independent MD trajectories with production runs of 1.0 μs and 0.5 μs each from uncorrelated starting structures. Both simulations show that the absence of the terminal Gal triggers a conformational change in the flexible C-C' loop, which can extend into the space that was occupied by the Gal in the complex with GM1, see Fig. 2i and j. This shift contributes to pulling the Ig V-set domain away from the membrane, lifting out of the bilayer the embedded Trp127 and Lys126 sidechains, ultimately compromising the stability of the complex. The change of the orientation of the Ig V-set domain in the complex with GM2 is highlighted by the analysis of the tilt angle in Fig. 2k. Because GM3 is shorter than GM2, missing the terminal Gal-β(1-3)-GalNAc relative to GM1, we expect the same outcome.

## Siglec-6 binding of ganglioside-enriched liposomes

As the MD simulations predicted Lys126 and Trp127 in mediating Siglec-6-ganglioside binding when the ganglioside is presented from a lipid bilayer, we created point K126A and W127A mutations and stably expressed these in CHO cells. Results are shown in Fig. 3. Additionally, as we had previously demonstrated that Arg122 was not required for Siglec-6 to engage with sialosides presented from a bilayer[14], we posited that another positively charged residue in the V-set may compensate in the absence of Arg122. We analysed the binding pocket of Siglec-6 and found that Lys129 is in relatively close spatial proximity to Arg122 and hypothesised that Lys129 may be able to compensate in the absences of Arg122. Expression of the Siglec-6 mutants were validated by flow cytometry to be expressed at similar levels as WT Siglec-6, see Fig. 3a. Therefore, we tested these Siglec-6-expressing cells for their ability to engage fluorescent liposomes bearing a neoglycolipid that we previously optimised as a Siglec-6 ligand[14,31], see Fig. 3b. In this assay, two controls were used: untransfected CHO cells and liposomes lacking the neoglycolipid. We observed a modest 25% reduction, which was statistically significant, in liposome binding to R122A, K126A, and K129A Siglec-6 cells. In line with the results from our previous study, we observed that liposome binding was only modestly decreased (~75% of WT) upon the mutation of the canonical arginine residue (Arg122), a feat unique to Siglec-6. The double mutant of R122A and K129A Siglec-6 showed a greater reduction in binding compared to each single mutant (~50% of WT) but was still above background signal, suggesting that additional interactions contribute to recognition of GM1. However, in the W127A mutant, nearly all binding to the glycolipid liposomes was lost (~5% of WT). Moreover, a near complete loss in binding was observed to the K126A W127A double mutant. These results demonstrate that Trp127 is the major driving force for the recognition of glycolipids presented from a bilayer by Siglec-6.

To assess whether W127A Siglec-6 perturbs the structure of Siglec-6, we tested the ability of W127A Siglec-6 to bind to glycolipids outside a bilayer in an ELISA, using conditions that we have previously optimised[32,33], see Fig. 3c. We observed no significant difference in binding between WT and W127A Siglec-6 to a neoglycolipid ligand developed previously[14] to engage Siglec-6. Moreover, none of the full length Siglec-6 mutants showed significantly perturbed expression levels. These results strongly suggest that W127A Siglec-6 is functional, and that the loss of binding for W127A Siglec-6 observed in the cell assay is not due to structural changes in the protein linked to the mutation.

## Siglec-6 binding to GM1-3 as free oligosaccharides

The characterisation of the recognition and binding mechanism of gangliosides by Siglec-6 was extended to include native MS (nMS) binding assays with GM1, GM2, and GM3 as free oligosaccharides, i.e. GM1os, GM2os and GM2os. We used Concentration Independent (COIN)-Catch-and-Release (CaR)-nMS assay[26] to quantitatively screen free ganglioside oligosaccharides binders in unknown concentration conditions. This approach is based on the slow mixing of solutions inside a nanoESI emitter to achieve a nearly constant glycan concentration flux[26]. Results are shown in Fig. 4a for Siglec-6 WT and for the W127A mutant, and in Fig. 4b for the R122A mutant. Unlike the case of the membrane-linked gangliosides, Siglec-6 binds the monosialylated gangliosides GM1-3os with similarly low affinity. Binding of the free oligosaccharides is largely unaffected by the mutation of Trp127 (W127A), further suggesting that this mutation does not significantly perturb the structure of Siglec-6 unlike binding to glycolipids in a membrane. Binding is significantly weakened by the loss of the canonical Arg122. Taken together these results suggest that, unlike the case of membrane-linked gangliosides, binding of GM1-3os occurs primarily through the canonical Arg122. As an interesting point, the complex of CD33 with a sialoside analogue (PDB 7AW6), shows an alternative conformational arrangement of the glycan epitope relative to the conformation required for a membrane-bound ganglioside. Indeed, in this complex the reducing end (GM1-equivalent) Gal-β(1-4)-[2-aminoethyl]-Glc moiety is directed towards the C-C' loop of CD33, rather than in the opposite direction, see Fig. 4h.

## Siglec-6 binding to GM1-enriched lipid nanodiscs

We used Catch-and-Release nMS performed with Ion Mobility Separation (IMS-CaR-nMS) to assess the ability of Siglec-6 to bind phospholipids in addition to gangliosides. To this end, we performed experiments with Siglec-6-FcWT, and R122A and W127A mutants with a GM1 nanodisc (ND) prepared with 10% GM1 and DMPC. Shown in Fig. 4c and d are: representative IMS heat maps (m/z versus IMS drift time $t_D$) and mass spectrum measured in negative ion mode for 0.8 μM Siglec 6 WT and 5 μM 10% GM1/DMPC ND (corresponding to 100 μM GM1 and ~900 μM DMPC), respectively. Both Siglec-6 and ND appeared as broad peaks and centred at m/z 7000 (tD = 12.5 to 19.0 ms) and 9500 (tD = 22 to 32 ms), respectively. Due to the broad m/z distributions of both Siglec-6 and ND, the free and lipid-bound Siglec-6 ions and those of the ND presumably overlapped in the mass spectrum. To address this potential issue, we used quadrupole isolation (m/z 7000 ± 100) along with IMS to separate the Siglec-6 (and Siglec-6–lipid complexes) ions from the ND ions. Collision-induced dissociation (CID) was performed in the Transfer region (100 V collision energy) to release lipid ions exclusively bound to the protein. The IMS heat map of the IMS-CaR-nMS assay and the CID mass spectrum are shown in Fig. 4e and f, respectively. The CID shows the release of deprotonated GM1 ions, with deprotonated DMPC ions also detected at lower abundance. Taken together the IMS-CaR-nMS results support the binding of Siglec-6 WT to both GM1 and to phospholipids. We performed analogous binding assays on the Siglec-6 R122A and W127A mutants with GM1/DMPC ND. The CID mass spectra extracted from IMS are shown in Fig. 4g. The results indicate that the Siglec-6 R122A mutant binds phospholipids analogously as the WT, meanwhile Siglec-6 W127A produces no detectable lipid ions, which confirms that the loss of Trp127 compromises lipid binding.

## Siglec-6 binds liposomes and nanodisks devoid of GM1

As the MD simulations and the mutagenesis studies demonstrated, Trp127 interacts with the bilayer and is critical for the binding of membrane-bound glycolipids by Siglec-6. Therefore, we hypothesised that there is direct binding between the naked liposomes and Siglec-6 and used the cell-based assay to assess if this was the case. Indeed, WT Siglec-6+ cells showed significantly higher binding to naked liposomes compared to Siglec-6- CHO cells, see Fig. 5a. Moreover, the binding of naked liposomes was greatly decreased (~5-fold) in W127A Siglec-6 CHO cells. In line with the results from Fig. 3b, naked liposome binding was not reduced to background levels with W127A Siglec-6, suggesting that while Trp127 is the major contributor

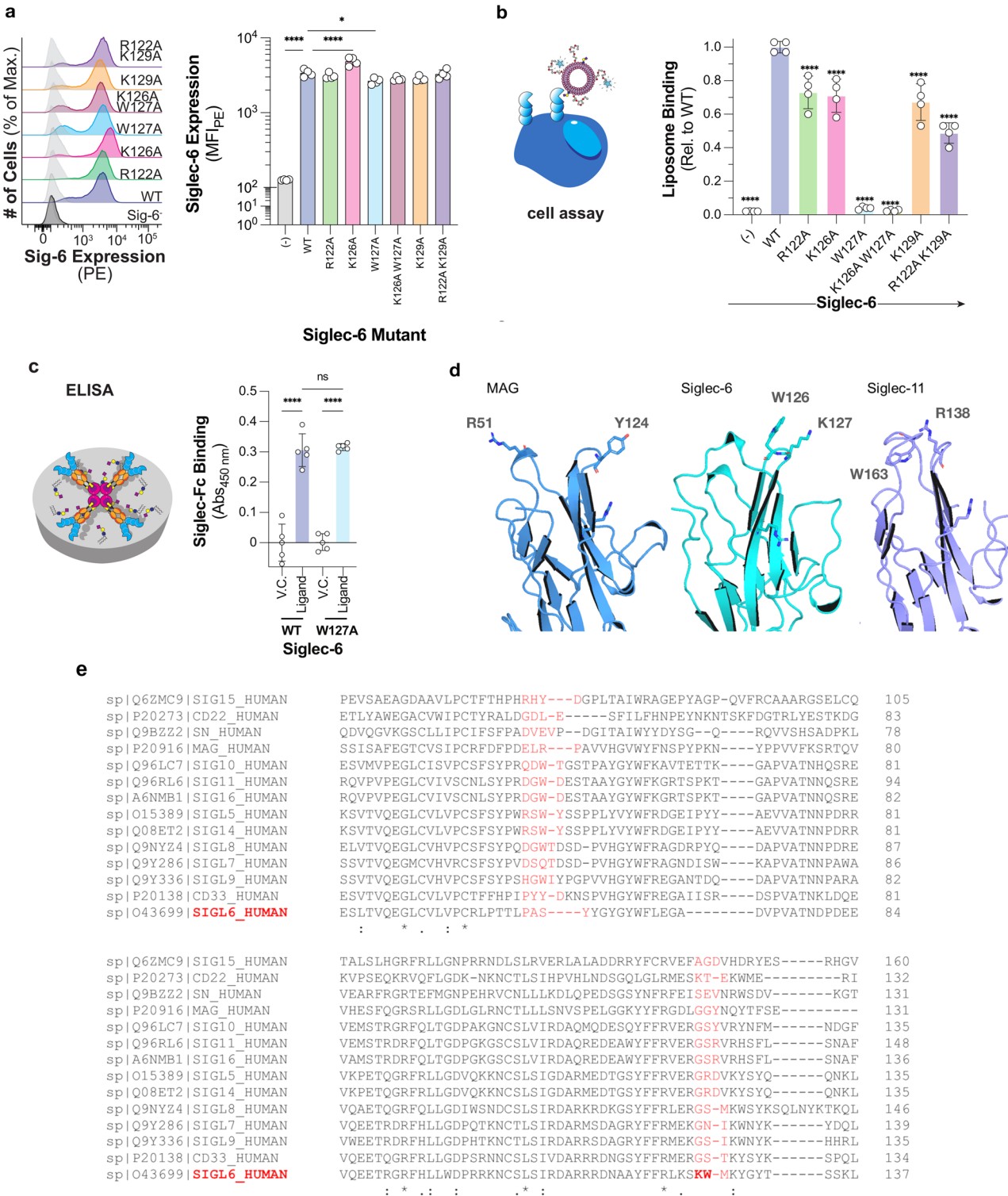

**Fig. 3 | Mutation of key residues W127 and K126, but not of the canonical R122, demonstrates loss of binding to gangliosides in liposomes by flow cytometry.** **a** Expression levels of the Siglec-6 WT and mutants obtained by flow cytometry ($n = 4$). Error bars correspond to standard deviation values in all panels. **b** Depiction and results of the cell assay used to assess the ability of Siglec-6 mutants to bind glycolipid liposomes using flow cytometry ($n = 4$). **c** Depiction and results of the ELISA approach used to assess the function of the W127A Siglec-6 Mutant ($n = 4$). Figures in panels a to c were generated using Adobe Illustrator and include only original elements. **d** 3D structures of the Ig V-set domains of MAG (blue cartoons; PDB 2ZG3), Siglec-6 (cyan cartoons; this work) and Siglec-11 (purple cartoons; AF-

Q96RL6-F1). The residues in the membrane-facing loops that could potentially interact with the bilayer are labelled and highlighted with sticks. Molecular rendering done with Visual Molecular Dynamics[55] (VMD; https://www.ks.uiuc.edu/Research/vmd/). **e** Sequence alignment of all human Siglecs performed with Clustal Omega[58] (https://www.ebi.ac.uk/jdispatcher/msa/clustalo) shows that only Siglec-6 has a KW (or similar) combination of residues on the same loop, yet a number of other Siglecs, namely 5, 8, 10, 11, 14 and 16, have a combination of one aromatic and one positively charged residues across the two loops facing the cell membrane when binding gangliosides.

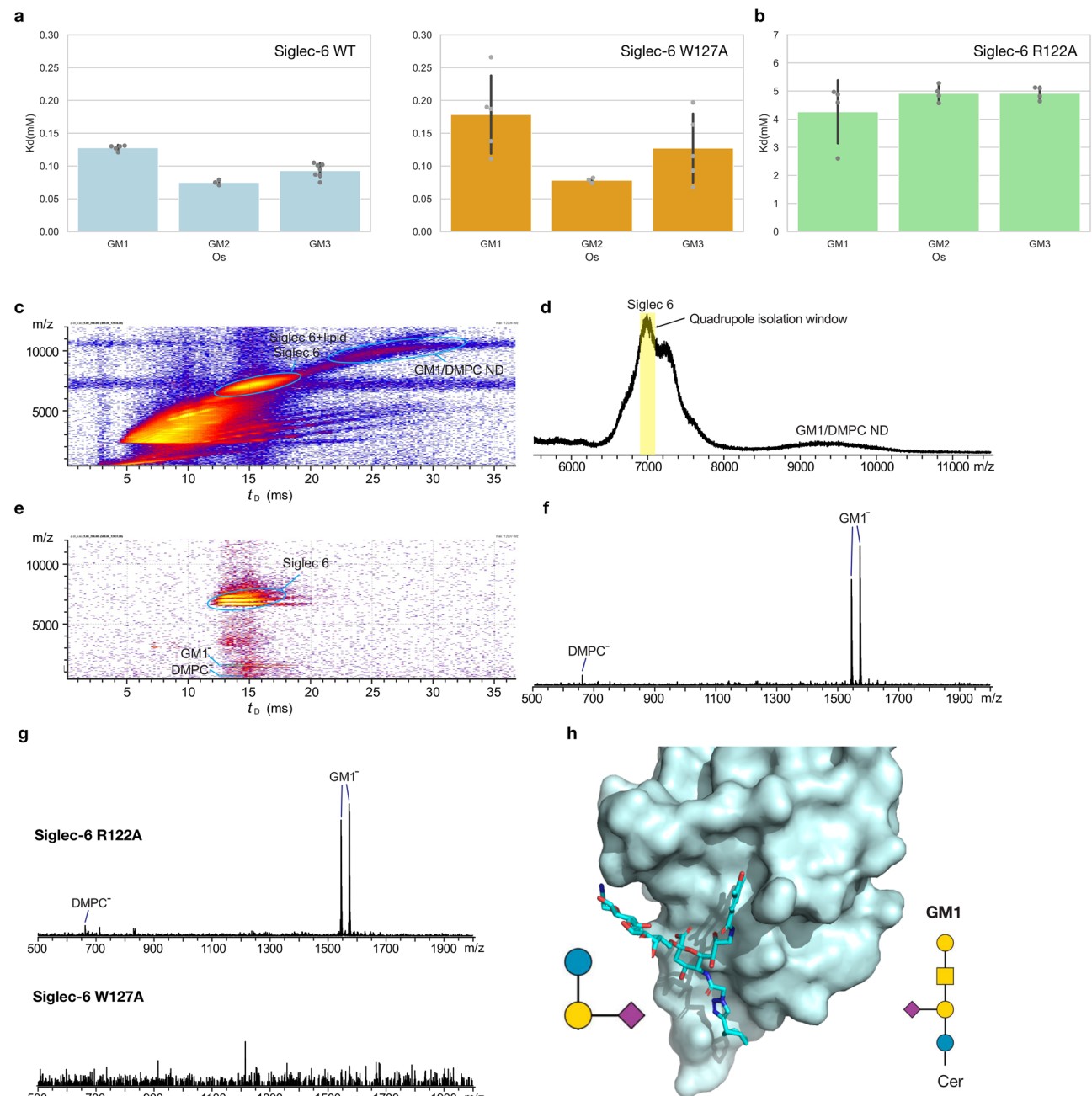

**Fig. 4 | Siglec-6 binds monosialylated gangliosides epitopes as free oligo-saccharides in a R122-dependent manner, with the same affinity. a** Binding affinities ($K$d, mM) of the ganglioside oligosaccharides GM1os, GM2os and GM3os for Siglec-6 (Fc) WT (light blue bars; $n_{(GM1os)} = 5$, $n_{(GM2os)} = 3$, $n_{(GM3os)} = 7$) and W127A mutant (orange bars; $n_{(GM1os)} = 5$, $n_{(GM2os)} = 3$, $n_{(GM3os)} = 5$) measured in aqueous ammonium acetate (200 mM, pH 6.8, 25 °C) by COIN-CaR-nMS. Bar plots generated with *seaborn* (https://seaborn.pydata.org/). Error bars correspond to standard deviationGL1 values in all panels. **b** Binding affinities ($K$d, mM) of the ganglioside oligosaccharides GM1os, GM2os and GM3os for Siglec-6 (Fc) R122A mutant (green bars; $n_{(GM1os)} = 4$, $n_{(GM2os)} = 4$, $n_{(GM3os)} = 4$) measured in aqueous ammonium acetate (200 mM, pH 6.8, 25 °C) by COIN-nMS. **c–g** IMS-CaR-nMS measurements performed in negative ion mode for 0.8 µM Siglec 6 (Fc) WT with 5 µM 10% GM1/DMPC ND in 200 mM aqueous ammonium acetate solution (pH 7.4). **c** IMS heat map (m/z versus IMS drift time $t_D$) and **d** corresponding full mass spectrum extracted from IMS heat map using Driftscope. **e** IMS heat map and **f** corresponding extracted CID mass spectrum showing (glyco)lipid ions released from Siglec-6; ions with m/z 7000 ± 100 were isolated by quadrupole (isolation window highlighted in yellow in **e** followed by IMS and CID; a collision energy of 100 V was applied in the Transfer region. **g** IMS-CaR-nMS measurements for Siglec-6 (Fc) mutants (each at 0.8 µM) and 10% GM1/DMPC ND (5 µM) in 200 mM aqueous ammonium acetate solutions (pH 7.4); CID mass spectra (extracted form IMS heat maps) showing (glyco)lipid ions released from (top) Siglec-6 R122A and (bottom) Siglec-6 W127A. **h** 3D structure of the V-set domain of CD33 in complex with a sialoside analogue (PDB 7AW6). The protein is rendered in cyan as a surface, while the sialoside is rendered with sticks with C atoms in cyan, O in red and N in blue. SNFG symbols of the sialoside (left) and of the GM1 (right) are shown to indicate the position of the glucose at the reducing end.

to the bilayer binding, other residues such as K126A also support the interaction. In support of the results obtained on naked liposomes, we also performed IMS-CaR-nMS measurements to test lipid binding of Siglec-6 WT and mutants on 'empty' DMPC ND (5 µM), see Fig. 5b. The CID mass spectra show that Siglec-6 WT binds phospholipids and that the R122A mutant retains lipid binding. Results obtained for the Siglec-6 W127A mutant demonstrate that the loss of Trp127 compromises the interaction with the bilayer.

**Fig. 5 | Siglec-6 interacts with the lipid bilayer in naked liposomes and nanodisks. a** Binding of naked liposomes (liposomes that lack a ligand) to WT, W127A and Siglec-6 CHO cells using the cell assay (*n* = 4). Error bars correspond to standard deviation values. Figure generated using Adobe Illustrator. **b** IMS-CaR-nMS measurements for Siglec-6 (Fc) WT and mutants (each at 0.8 μM) with DMPC ND (5 μM) in 200 mM aqueous ammonium acetate solutions (pH 7.4); CID mass spectra (extracted form IMS heat maps) showing lipid ions released from (top) Siglec-6 WT, (middle) Siglec-6 R122A and (bottom) Siglec-6 W127A.

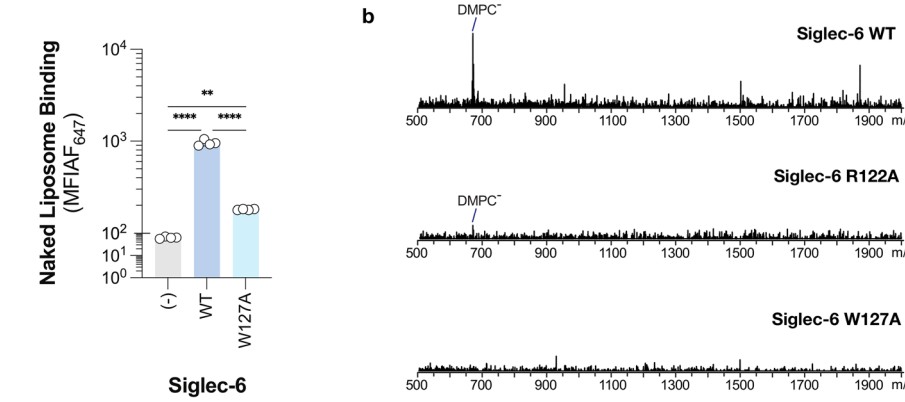

## Discussion

Characterising glycan-binding specificity across human Siglecs is a difficult task, hindered by their weak binding affinity, potential promiscuity, and structural plasticity facilitating both *cis* and *trans* ligand interactions. Identifying the molecular determinants regulating their glycan preference is even more complex, as it requires gathering statistically reproducible, atomistic-level information on the structure and dynamics of the Siglecs, integrated within a sufficiently detailed 3D model of their own biological environment. In this study, we show how all-atom classical MD simulations were used to guide experiments towards the characterisation of the unique binding mechanism and glycan-specificity of human Siglec-6, an inhibitory receptor expressed in mast cells, memory B-cells and, uniquely to humans[34], placental syncytiotrophoblasts.

Our results show that Siglec-6 evolved the ability of recognising and binding monosialylated gangliosides not only by engaging the glycan epitope poking out of the bilayer, but also by supplementing its binding affinity through interactions with the surrounding phospholipids. As suggested by MD simulations and supported by liposome and ND binding assays, the interactions between the Siglec-6 and the cell membrane hinge on the insertion of Trp127 into the bilayer, assisted by the adjacent Lys126, which interacts with the phospholipid heads. Different examples of proteins anchoring phospholipid bilayers through exposed aromatic and charged amino acids have been described within various biological contexts, e.g. where such interactions are instrumental for the orientation of transmembrane proteins[35–38] within the bilayer, as well as for the interaction of proteins and peptides with the membrane[39–41] and for the binding of membrane-embedded ligands[40,42–44]. To our knowledge, this is the first instance in which binding to the cell membrane has been characterised as a key step in the ligand-recognition of a carbohydrate-binding protein.

Further to this, we find that the structure of the ganglioside epitope is a determinant for Siglec-6 recognition and binding, with GM1 being the only monosialylated lipid-linked ganglioside recognised within a phospholipid bilayer. As in other Siglecs[1], Siglec-6 binding specificity is controlled by the C-C' loop, which interacts with the terminal Gal in GM1. GM2 and GM3 lack this terminal Gal and Gal-β(1-3)-GalNAc motif, respectively. In a model of the Siglec-6/GM2 complex that we studied by two independent MD simulations, we see that the C-C' loop is conformationally dynamic, enabled by the absence of the terminal Gal, triggers the disruption of the contact with the membrane and ultimately promoting the dissociation of the complex. Mutagenesis and binding assays on GM1-enriched liposomes and ND confirm that bilayer interactions mediated by Siglec-6 requires Trp127, with the support of the adjacent Lys126. As demonstrated through binding assays on 'empty' liposomes and ND, the interaction with the bilayer persists even in the absence of gangliosides. These results inform an interesting hypothesis about the ganglioside recognition strategy, whereby Siglec-6 scans the cell membrane surface through low affinity interactions with lipids, in search for higher affinity and specificity binding epitopes, such

as GM1. The MD simulations of isolated GM1 and GM3 gangliosides in a model bilayer suggest that such a strategy may be the most efficient process. Indeed, while we measured that the sialic acid in both GM1 and GM3 remains accessible through the lipids, the size of the exposed epitope is rather small, making it unlikely to be able to engage the Siglec-6 Ig V-set unless in very close proximity. To this end, the evolutionary advantage of developing a mechanism that hinges on membrane interaction could rest in enhancing Siglec-6 selectivity for GM1, namely turning a sialic acid-binding lectin, into a high-specificity molecular precision tool. The study of how such sophisticated scanning mechanism impacts epitope recognition and binding at the molecular level of detail would be very interesting, yet we believe to be beyond the scope of this work, and it may require the use of a coarse-grained MD approach, ideally in combination with super-resolution microscopy[45,46].

Siglec-6 binding has been shown to be independent of the canonical Arg[14], while MAG and Siglec-11 have been also reported to share such feature[15]. In agreement, our MD simulations show that the interaction between Siglec-6 and GM1 is only partially supported by the salt bridge between Arg122 and the Neu5Ac, with binding assays and mutagenesis in this and in earlier work[14] showing a reduction of the binding affinity in the R122A mutant. Sequence alignment shows that the KW motif implicated in the membrane interaction is a unique feature of Siglec-6, see Fig. 3e. Yet, structural analysis supported by the insight we gathered from the MD simulations, show that contacts with the membrane could potentially involve two loops, see Fig. 3d. Other Siglecs, namely 5, 8, 10, 11, 14, and 16, carry at least one aromatic and one positively charged residue (at physiological pH) on either loop. Although we cannot exclude that membrane interactions may occur in other Siglecs through this alternative 'two-loop' strategy, some of us showed through MD simulations that Siglec-10, which has a particularly extensive combinations of 'ideal' membrane-binding residues across the two loops, namely Arg47 and Trp50 on one, and Arg122 and Tyr125 on the other, see Fig. 3e, does not engage with the lipids in the bilayer[47]. This is not an entirely surprising result, as Siglec-10 does not need to bind the cell membrane because its binding specificity is for polysialylated epitopes in glycoproteins or on secreted glycans[47–50]. Siglec-11 does not need to bind the membrane either, as it does not bind gangliosides[14] but α(2-8) polysialic acids, which have a rigid architecture and terminate the arms of *N*-glycans[51]. Earlier work[14] indicates that MAG binds GM1 only weakly, but not GM2/3, and that it has a higher affinity for polysialylated gangliosides; this interaction appears to be dependent on its canonical Arg118. Ultimately, membrane binding appears, thus far, to be a feature unique to Siglec-6. Yet, the broad accessibility of the binding site in the terminal Ig V-set domain and the conservation of the salt-bridging canonical Arg, indicates that Siglecs can bind free sialosides within a varying range of low binding affinities. Indeed, contrary to the case of membrane-bound gangliosides, here we have shown through nMS binding assays that Siglec-6 binds GM1-3 as free oligosaccharides with similarly low binding affinities with a mechanism that is entirely dependent on the canonical Arg122. Based on

the crystal structure of the highly homologous Siglec-3 in complex with a sialoside analogue (PDB 7AW6), the relative orientation of the bound glycan epitope could also change to complement different (less constrained) environmental conditions.

Ultimately, the analysis of the results from MD simulations, binding assays and mutagenesis we presented in this work contribute to substantiate a unique and distinctive binding mechanism adopted by Siglec-6, supporting the evolution of distinct human Siglecs as molecular precision tools for the recognition of specific sialosides in their biological environment. Further studies in this area are currently in progress.

## Materials and methods
### Computational methods
**MD simulations of isolated GM1 and GM3 in a lipid bilayer**. Two independent systems were prepared, each embedding either a GM1 and GM3 ganglioside within a symmetric 130 Å × 130 Å lipid bilayer composed of 60% 1,2-distearoyl-sn-glycero-3-phosphocholine (DSPC) and 40% cholesterol (CHL1), using the CHARMM-GUI Membrane Builder tool[52]. All MD simulations were performed with AMBER18, using the CHARMM36m force field to parameterise the gangliosides and lipids. Each system underwent energy minimisation for 5000 steps (2500 steps of steepest descent followed by 2500 steps of conjugate gradient minimisation). Equilibration followed the six-step CHARMM-GUI protocol[53], during which positional restraints of 10 kcal/mol Å² were initially applied and then gradually reduced to 5 kcal/mol Å² for the protein and 2.5 kcal/mol Å² for the membrane, before being completely removed. Temperature was maintained at 315.15 K using Langevin dynamics ($\gamma\_ln$ = 1.0 ps⁻¹), and pressure was controlled semi-isotropically at 1 atm using the Berendsen barostat. Periodic boundary conditions were applied throughout. Long-range electrostatics were treated using the Particle Mesh Ewald (PME) method with an 11 Å cutoff. Bond lengths involving hydrogen atoms were constrained using the SHAKE algorithm, enabling a 2 fs integration time step. Each system was simulated for 1 μs. The orientation of the sialic acid headgroup (Neu5Ac) was quantified by calculating the tilt angle (θ), defined as the angle between the vector connecting atom C1 of the terminal galactose and atoms C2 and C3 of the Neu5Ac residue, relative to the axis perpendicular to the membrane plane. Tilt angle distributions were computed over the entire MD trajectories for both GM1 and GM3.

**MD simulations of GM1 and GM2 in complex with Siglec-6**. The Siglec-6/GM1 complex was constructed by aligning the conformation of GM1 from the simulation of the isolated glycolipids, see above, to the sialic acid moiety in the crystal structure of Siglec-3 bound to a sialoside analogue (PDB: 7AW6). The structure of Siglec-6 (AF-O43699) was validated by structural alignment with Siglec-3, resulting in a backbone RMSD of 0.704 Å. A second system, containing GM2, was generated by removing the terminal Gal residue from GM1. Both complexes were embedded in the same DSPC/cholesterol bilayer described above, and the simulations were carried out using AMBER v.22[54], following the same multistep equilibration and production protocol. In the GM1 system, a distance restraint (5 kcal/mol Å²) was applied between the side chain of Arg122 and the carboxyl group of the Neu5Ac to stabilise the salt bridge during the initial 400 ns of the 2.5 μs production simulation. The same approach was used for the GM2 complex; in both 1-μs replicas, removal of the restraint led to destabilization of the complex, indicating a loss of stable interaction between Siglec-6 and GM2 in the absence of the terminal Gal residue. All MD trajectories were analysed using Python3 scripts written in-house. Distances, occupancies, and tilt angles were calculated frame-by-frame using the *cpptraj* module in AMBER v.22 and the graphical user interface VMD[55]. Kernel Density Estimates (KDE) and all other plots were generated using the *matplotlib* (https://matplotlib.org/) and *seaborn* (https://seaborn.pydata.org/) libraries.

## Experimental methods
**Nanodisc preparation**. Nanodiscs (NDs), consisting of 10% GM1 and DMPC, were prepared using the protocol described by Sligar and coworkers[12]. Briefly, the lipids were diluted in methanol at the desired molar ratios, dried under gentle vacuum to form a lipid film and then resuspended in a buffer (pH 7.4) containing 20 mM TrisHCl, 0.5 mM EDTA, 100 mM NaCl and 25 mM sodium cholate (Sigma-Aldrich Canada, Oakville, Canada). The membrane scaffold protein MSP1E1 was added to the mixture at a MSP1E1:(GM1 + DMPC) molar ratio of 1:100. The ND self-assembly process was initiated by adding pre-washed biobeads (Bio-Rad, Mississauga, Canada) and the mixture was incubating at room temperature overnight on an orbital shaker. After incubation, the supernatant was recovered and the NDs purified using a Superdex 200 10/300 size-exclusion column (GE-Healthcare Life Sciences, Piscataway, NJ) equilibrated with 200 mM ammonium acetate (pH 7.4). Finally, the ND fraction was collected, concentrated and dialysed into 200 mM ammonium acetate (pH 7.4) using an Amicon microconcentrator (EMD Millipore, Billerica, MA) with a 30 kDa MW cutoff. All ND stock solutions were stored at –80 °C prior to analysis. Each ND sample consists of two copies of MSP1E1 and ~200 lipids. Therefore, ND concentration was estimated based on the UV absorption of MSP1E1 at 280 nm with the extinction coefficient of $\varepsilon_{280nm,MSP1E1}$ = 32,430 cm⁻¹ M⁻¹, and assuming [ND] = 1/2×[MSP1E1]. Each ND approximately contained 20 GM1 molecules and 180 DMPC molecules.

**Concentration independent native mass spectrometry with catch-and-release**. The Concentration Independent native mass spectrometry (COIN-nMS) assay, performed with Catch-and-Release (CaR), was implemented in negative ion mode using a Q-Exactive Ultra-High Mass Range (UHMR) Orbitrap mass spectrometer (Thermo Fisher Scientific, Bremen, Germany) equipped with a modified nanoflow electrospray ionisation (nanoESI) source, as described elsewhere[26]. In short, the nanoESI emitter was loaded with 2 solutions, solution 1 contains ammonium acetate (200 mM, pH 6.8) of Siglec-6 Fc (WT or mutants, 0.8 μM) and 0–0.5 μM of glycan of interest, solution 2 contains protein (at an identical concentration as in solution 1) and glycans of interest (30–40 μM). To perform nanoESI, a voltage of approximately −0.7 kV was applied to a platinum wire inserted inside the nanoESI tip and in contact with the solution. The solution temperature was 25 °C. Resolution (resolving power) of 25,000 was used. Maximum injection time was 200 ms, the S-lens RF level was 200 and DC offset was 21. Collision energy was 120 V. Raw data were processed using the Thermo Xcalibur 4.4 software. Time-resolved mass spectra were averaged over 1 min intervals and the sum of the charge state-normalised abundances of the reactant and the complex ions were calculated automatically using the SWARM software (https://github.com/pkitov/CUPRA-SWARM)[56]. The $K_d$ values were fit using Igor pro (WaveMetrics Inc., Lake Oswego, OR, USA) using Eq. 1:

$$F_t = DE \frac{[P]_0 + C_L t + K_d - \sqrt{\left(K_d - C_L t + [P]_0\right)^2 + 4K_d C_L t}}{2[P]_0} \quad (1)$$

where *DE* is the detection efficiency of the released glycan relative to the Siglec-6 Fc (P), $C_L(t)$ is the *t*-dependent function that describes the change in ligand concentration due to diffusion and advection, $[P]_0$ is initial protein (Siglec-6 Fc) concentration, the time-dependent fractional binding site occupancy (fraction bound, $F_t$) of P, was calculated using the time-dependent abundance ($Ab_t$) of the released ligand and free protein as shown in Eq. 2,

$$F_t = \frac{Ab_t(L)}{Ab_t(P)} \quad (2)$$

**Catch-and-release native mass spectrometry with ion mobility separation**. CaR-nMS, performed with Ion Mobility Separation (IMS),

was implemented in negative ion mode using a Waters Synapt G2S quadrupole-ion mobility separation-time of flight (Q-IMS-TOF) mass spectrometer (Waters, Manchester, UK) equipped with a NanoLock-Spray ion source. A nanoESI voltage of −0.8 kV, a source temperature of 80 °C and a cone voltage of 70 V were used. Argon was used in the Trap and Transfer ion guides at pressures of $2.77 \times 10^{-2}$ and $2.84 \times 10^{-2}$ mbar, respectively, with the Trap and Transfer voltages of 5 and 2 V, respectively. For IMS, a wave height of 40 V and a wave velocity of 650 m s$^{-1}$ were applied along with a helium and nitrogen (IMS gas) gas flow of 120 and 90 mL min$^{-1}$, respectively. Collision-induced dissociation (CID) by isolating ions with m/z 7000 ± 100 in the quadrupole, subjecting them to IMS, and then collisional activation using a 100 V collision energy in the Transfer region[57]. All data was processed using MassLynx software (v4.1) in combination with DriftScope v2.5.

**Mutagenesis**. Mutagenesis was achieved by three consecutive polymerase chain reactions using gene-overlap extension mutagenesis. The first reaction used a forward primer at the start of the gene containing a 5' *NheI* site and reverse primer that was centred on the mutation site and the complete gene of interest at the template. The second reaction used a forward prime that was centred on the mutation site and a reverse primer at the end of the gene that featured and 3' *AgeI* site and used the gene of interest as the template. The third reaction used the products of the first two PCR reactions as a template and the primers at the start and end of the gene. Following size validation using agarose gel electrophoresis, the final PCR products were digested with *NheI* and *AgeI* and then ligated into pCDNA5. Following ligation, the ligation product was then transformed into *E. coli* DH5α. Colonies were then grown in LB overnight at 37 °C and then miniprepped. The minipreps were then validated by restriction digest and Sanger sequencing.

**Cell culture**. Flp-In Chinese hamster ovary (CHO) cells were cultured as previously described[14], but in short, CHO cells were cultured in DMEM/F12 Media (Gibco) supplemented with 5% (V/V) foetal bovine serum (Gibco), penicillin (100 U/mL), and streptomycin (100 μg/mL). Cells were grown at 37 °C and 5% CO$_2$ in tissue culture dishes.

**Stable transfection of Flp-In CHO Cells**. CHO Flp-In cells were initially cultured as described above and in ref. 14. The transfection follows the same protocol described in ref. 14, and began by seeding 400,000 cells in a six-well cell culture dish. The next day, 0.2 μg of the desired Siglec DNA in pCDNA, 2 μg of pOG44 plasmid, and 7 μg of Lipofectamine Plus reagent (Thermo Fisher) were added to 250 μL of Opti-MEM (Gibco) and the mixture was incubated at room temperature for 15 min. Next, 8 μL of Lipofectamine LTX reagent (Thermo Fisher) was added to the mixture; and the mixture was left at room temperature for 30 min. During the incubation, the seeded cells were gently washed with Opti-MEM. The transfection mixture was then added to the seeded cells, and the cells were left in the growing conditions described above overnight. The next day the media was aspirated from the well and replaced with CHO growth media, as described above and in ref. 14. The cells were then selected over 2 weeks by gradually increasing the amount of Hygromycin B from 0.5 to 1 mg/mL, replacing the media every other day.

**Liposomes preparation**. Liposomes were prepared by hydrating lipid thin consisting of DSPC (61.5 mol%), cholesterol (38 mol%), and PEG$_{45}$-DSPE (0.4 mol) and AF647-PEG$_{45}$-DSPE films with PBS pH 7.4. Glycolipids were added at the expense of DSPC. The lipid solutions were extruded using an Avanti-mini extruder using 800 nm and 100 nm filters respectively. Liposomes were then validated with dynamic light scattering. More details regarding liposome production can be found elsewhere[14,31].

**Siglec-Fc expression and purification**. Siglec-6 Fc constructs were expressed and purified as described elsewhere (PMID: 37149531). Briefly,

Siglec-Fcs were expressed from Chinese hamster ovary (CHO) Flp-In® cells. CHO cells stably transfected with Siglec-6 Fcs constructs were culture in 1% (V:V) FBS and penicillin (100 U/mL), and streptomycin (100 μg/mL) for ten days post confluence. Following expression, the media was collected, centrifuged at $300 \times g$, sterile filtered and stored at 4 °C. The Siglec-Fc was isolated from the media via affinity chromatography. First, the Siglec-Fc was purified using a HisTrap™ excel (Cytiva) and eluted using Imidazole buffer (20 mM sodium phosphate, 500 mM sodium chloride, 500 mM imidazole, pH 7.4) and collected in 1 mL fractions. Fractions were checked for protein via nanodrop A$_{280\ nm}$ and fractions that had protein were combined and diluted 5-fold in Buffer W (100 mM Tris–HCl, 150 mM sodium chloride, 1 mM EDTA, pH 8). The Siglec-Fc was then loaded onto a Strep-Tactin® Superflow® (IBA) column and eluted using Buffer E (100 mM Tris–HCl, 150 mM sodium chloride, 1 mM EDTA, 5 mM desthiobiotin, pH 8). Fractions containing protein, determined by A$_{280}$ nm, were combined and dialysed against PBS for 24 h at 4 °C. Dialysed Siglec-Fc proteins were concentrated using a 50 kDa ultra-centrifugal device (Amicon) to a concentration of 0.1 mg/mL. The Siglec-Fc proteins were aliquoted, frozen at −80 °C, lyophilised, and stored at −20 °C. Siglec-Fc proteins were then validated by SDS-PAGE and BCA.

**Cell lines**. CHO Flp-In cells were purchased from ATTC.

**Cell assay**. 150,000 cells were added to a 96 well U-bottom microplate. Liposome solutions were prepared at 50 μM in 1% (g:mL) BSA PBS pH 7.4. 50 μL of liposome solution was added to the cells and the cells were incubated with the liposomes for 30 min at 37 °C. The cells were then washed with PBS twice. Following the washes, the cells were then resuspended in anti-Siglec antibody solution (1/250 V:V antibody in 500 μM EDTA, 1% BSA (g:mL)). The cells were then incubated at 4 °C for 30 min. The cells were again washed and then analyzed by flow cytometry. Antibody information can be found in Table S.2, and general gating strategy can be found in Fig. S3.

The chemical structure of neoglycolipid ligand (NGL1) used in this experiment is shown below:

The chemical synthesis of NGL1 was originally described in ref. 14 and is reported as follows for completeness. To a stirred solution of trisaccharide azide (8.0 mg, 11.0 μmol) and alkyne (15.0 mg, 26.0 μmol) in a mixture of THF (3 mL) and water (3 mL) at room temperature were added N,N-diisopropylethylamine (6.0 μL, 34.0 μmol), copper (II) sulfate pentahydrate (53.0 mg, 212.0 μmol) and L-ascorbic acid sodium salt (81.0 mg, 408.0 μmol) successively. The reaction mixture was shielded from light (aluminum foil) and stirred overnight. The reaction mixture was then concentrated, and the residue was purified by size exclusion column chromatography (Sephadex-LH-20, CH2Cl2CH3OH, 1:1) to afford 1 (10.0 mg, 69%). Rf 0.3, EtOAc–CH3OH–HOAc–H2O (36:9:9:6); 1H NMR (700 MHz, CD3OD plus a few drops of CDCl3, H) 8.03 (s, 1H), 4.65–4.55 (m, 2H), 4.39 (d, 1H, J = 7.0 Hz, H-1), 4.31 (d, 1H, J = 7.8 Hz, H-1), 4.24–4.20 (m, 2H), 3.95–3.40 (m, 28H), 3.22–3.20 (m, 1H), 2.90–2.80 (m, 1H), 2.10–2.00 (m, 2H), 1.99 (s, 3H, NHCOCH3), 1.70–1.50 (m, 5H), 1.40–1.20 (m, 56H), 0.86 (dd, 6H, J = 6.9 Hz); HRMS (ESI) calcd. for [M − H]− C63H116N4O22 1279.8008, found 1279.8014.

**Flow cytometry**. Flow cytometry measurements were collected on a 5-laser Fortessa X-20 (BD Bioscience). All the resulting data were analysed using FlowJo (10.5.3) software (BD Biosciences)

**Siglec-Fc ELISA**. A detailed description of the approach can be found elsewhere[14,31]. In short, 50 μL of 50 μM glycolipid solution was transferred to a 96 well ELISA microplate. The plate was left at 37 °C for 2 to remove the ethanol. The wells were blocked with 5% (g:mL) bovin serum albumin (BSA) for 1 h at room temperature. During the blocking step, the Siglec-Fc-Strep-Tactin solution (Siglec-Fc solution: 2 μg/mL, Strep-Tactin-HRP: 0.13 μg/mL-IBA) was prepared and complexed at room temperature for at least 30 min at room temperature. The blocking buffer was then removed, and the Siglec-Fc solution was added for 2 h at room temperature. Unbound Siglec-Fc solution was then removed, and the amount of binding was measured using TMB substrate. The reaction was then quenched using 1 M $H_3PO_4$ and the $Abs_{450}$ was measured using a Molecular Devices SpectraMAX iD5.

## Statistics and reproducibility

Statistical analysis of the simulation data analysis is based on kernel density estimate (KDE) where maxima values are stated in captions, see Fig. 2. Other analysis consists in time evolution and counts of specific parameters, namely hydrogen bonding distances and insertion of residues through a membrane estimated along single trajectories. All binding assays have been performed in replicates with $n \geq 3$, see Figs. 4 to 5. Data points are shown complemented by error bars corresponding to standard deviation values.

## Reporting summary

Further information on research design is available in the Nature Portfolio Reporting Summary linked to this article.

## Data availability

All MD trajectories and related structures and data are available open access at https://glycoshape.org/downloads/Siglec-6, at https://glycoshape.org/, and at https://zenodo.org/records/17312619. Data used for the statistical analysis of the MD simulations are available in https://doi.org/10.5281/zenodo.17877366. The raw native MS dataset is available on the MassIVE database (MSV000100484) https://doi.org/10.25345/C5028PS8P. Data points from experiments are available as Supplementary Data. All other information in available in Supplementary Material.

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

## Acknowledgements
IT Solutions at the University of Southampton is gratefully acknowledged for the generous allocation of computational resources on the HPC cluster Iridis. The Science Foundation of Ireland (SFI) Frontiers for the Future Programme is gratefully acknowledged for financial support. The opinions, findings, and conclusions or recommendations expressed in this material are those of the author(s) and do not necessarily reflect the views of the Science Foundation of Ireland. The following funding agencies are gratefully acknowledged for financial support: Science Foundation of Ireland (SFI) Frontiers for the Future grant 20/FFP-P/8809 (E.F.), NSERC grant RGPIN-2018-03815 (M.S.M.), GlycoNet grant CR-03 (M.S.M.), GlycoNet Integrated Services grant CFI-MSI (J.S.K.), Canada Excellence Research Chairs program (L.K.M.).

## Author contributions
Conceptualisation: E.F. and S.D.A.; Methodology: E.F., M.S.M., J.S.K., S.D.A., E.N.S., D.B., O.S., and L.H.; Investigation: E.F., M.S.M., J.S.K., S.D.A., E.N.S., D.B., O.S., and L.H.; Visualisation: E.F., M.S.M., J.S.K., S.D.A., E.N.S., D.B., O.S., and L.H.; Supervision: E.F., M.S.M., J.S.K. and L.K.M.; Writing of original draft: E.F., S.D.A., M.S.M, E.N.S., J.S.K., and D.B.; Final review and editing of the manuscript: E.F., S.D.A., O.S., M.S.M., E.N.S., J.S.K., L.K.M. and D.B.

## Competing interests
The authors declare no competing interests.
