## [Transparent Peer Review file · Communications Biology]

Glycolipid recognition and binding by Siglec-6 hinges on interactions with the cell membrane

Corresponding Author: Dr Elisa Fadda

Version 0:

Reviewer comments:

Reviewer #1

(Remarks to the Author)

Dear Dr. Fadda and co-workers,

I was very excited to read your manuscript on the surprising new binding mode of Siglec-6. I enjoyed the manuscript for its combination of computational structural biology, molecular dynamics simulations, and a state-of-the-art set of actual binding experiments. As the 'siglec-mystery' keeps prevailing in the carbohydrate community, I think this is a very important, timely, and thorough addition to this field and therefore can only recommend its publication.

This is not without some comments. These are minor, as this paper has clearly been through review already. They mainly center around one key question that is not mentioned in the discussion, and that is 'why has this binding mechanism evolved?'. What evolutionary reason could there be for evolving a binding mechanism that involves the membrane? I think including a few lines of text on what the author's take on this is, would be very nice.

The other minor point was that, for clarity, the authors include one or two lines more of text on their mass spec methods in the main text. I had to now look it up via the reference, which many readers may not want to do.

Final point. Page 3, line 89 misses a space between the reference (15-18) and the word 'can'.

Reviewer #2

(Remarks to the Author)

Siglecs are immunoregulatory receptors that recognise sialylated glycoconjugates. In most of the 14 human Siglecs, binding of sialic acid involves a conserved arginine in the V-set domain. Here, using all-atom molecular dynamics (MD), mutagenesis, and binding assays with liposomes and nanodiscs, we show that Siglec-6 engages membrane-embedded GM1 through a mechanism that only partly depends on this canonical Arg122. MD reveals insertion of Trp127 into the lipid bilayer and contacts of Lys126 with phospholipid headgroups that supplement binding to the GM1 epitope; disrupting these residues markedly reduces binding to GM1 presented from membranes. In contrast, binding to free GM1/GM2/GM3 oligosaccharides is weakened by R122A and is insensitive to W127A, indicating Arg122-dependent recognition off-membrane. The data support a model in which Siglec-6 samples membranes via low-affinity lipid interactions and achieves specificity for GM1 when its terminal galactose engages the C-C' loop, allowing intermittent Arg122-Neu5Ac contacts. This work clarifies how Siglec-6 adapts sialoside recognition to the membrane environment.

Overall, the manuscript is very well written and presents high-quality data. However, I have some (minor) points that should be clarified before publication:

1. The proposed mechanism depends on the V-set C-C' loop (and Lys126/Trp127) geometry. AF loop confidence (pLDDT) is often low for exposed loops. It would be helpful to show the per-residue AF confidence for the C-C' loop and the KW segment, perhaps in a supplementary figure.
2. Please provide a rationale for the chosen membrane composition used in the MD simulations, nanodisc, and liposome studies. It would also be useful to know whether Trp127 insertion and Arg122 intermittency persist with alternative membrane compositions, such as POPC:chol with or without sphingomyelin.

3. The complex was seeded by aligning membrane GM1 to a sialoside analogue pose from Siglec-3 (PDB 7AW6), then aligning Siglec-6 to Siglec-3. Could the authors clarify whether the observed ligand orientation is driven by the membrane and KW contacts rather than inherited from the initial alignment?
4. Fig. 2f/h qualitatively show intermittent contact defined with a 5 Å threshold. Please report occupancy (% time, mean lifetimes, and transition counts (\pm SEM across replicates) for the Arg122–Neu5Ac salt bridge. And also provide a free-energy proxy, e.g., potential of mean force (distance CV) or alchemical $\Delta\Delta G$ (R122A vs WT), to quantify how “dispensable” Arg122 is in membranes vs free oligosaccharides.
5. The authors simulate both non-glycosylated and fully glycosylated Siglec-6 but primarily report the former. Given the KW segment is near the V-set domain, N103 (the only V-set sequon) could influence loop dynamics. Please summarise whether the fully glycosylated model preserves KW membrane insertion and Arg intermittency. If possible, testing an N103Q mutant could clarify whether V-set glycosylation modulates GM1 binding.
6. Minor typo (p. 10): “GM1-3os ... GM2os and GM2os” should read GM1os, GM2os, GM3os.

Reviewer #3

(Remarks to the Author)

Before starting my review, I would like to mention that my expertise lies primarily into molecular dynamics simulations and protein-lipid or protein-sugar interactions. My knowledge is more limited in the areas of mass spectrometry and cell biology. So, I will focus my review on the theoretical results and will not assess the methodological aspects of the experimental work.

Overall, this is an interesting manuscript that combine Molecular Dynamics (MD) simulations and biochemical assays to shed new lights on lectin-glycolipid interactions. Specifically, S. D’Andrea et al. identified new residues (K126 and W127) at the surface of Siglec-6 involved in direct interactions with phospholipids. The biochemical assays coupled to mutagenesis appear to be in good agreement with theoretical results. However, the MD analyses would benefit from a more extensive and quantitative treatment of dynamic processes observed throughout the simulations, rather than focusing on the final frames. This would significantly strengthen the theoretical part of the study. In addition, placing the work in the broader context of previously published MD studies (both atomistic and coarse-grained) in the Introduction and Discussion would help readers appreciate what can currently be achieved using MD simulations.

Major points

- 1- In introduction, a paragraph summarizing theoretical works (and especially MD simulations) on protein-sugar or protein-lipid interactions would help the readers to understand what it is possible to do with this type of approach.
- 2- The authors should clarify the source of the lipid composition used for the membrane (60% DSPC + 40% cholesterol) and how it relates to biological membranes. From my experience, cellular membranes are often asymmetric (outer and inner leaflets differ in composition), which is not the case here. Why was asymmetry not considered ? With such a high cholesterol concentration, the membrane may be in a gel phase (see Fig. 2a and 2g). Is this a behavior the authors intended to reproduce ? How might this affect GM1 sugar moiety exposure and Siglec-6 interactions with the membrane ? All these points need, at least, to be discussed.
- 3- How the construction of the system can influence the final results ? In particular, did the authors consider using coarse-grained MD simulations to study the full process of interaction of siglec-6 with the membrane by placing the protein away from the membrane and allowing it to diffuse. All the models to perform such simulations are readily available, and computationally less expensive than atomistic MD simulations. This should, at least, be discussed.
- 4- The distance analysis presented in Fig. 2-f is quite a simple proxy for identifying siglec-6 interactions with the membrane. Can a slightly more sophisticated and exhaustive analysis provide additional details on which surface residues, beyond the ones described by the authors, are involved ? Tools such as MDAnalysis can perform facilitate such analyses. Furthermore, this type of analysis should also to be performed for siglec-6 in interaction in GM1, as presented in fig. 2-a.
- 5- Analysing siglec-6 tilting (has done for GM1) in interaction with GM1 and GM2 can help assess the flexibility, the dynamic, and the stability of the system.
- 6- A proper analyse of hydrogen bonds and salt bridge stability over time is necessary to quantify the interactions of siglec-6 with sugar moieties and lipids. MDAnalysis or VMD can easily perform such analysis.
- 7- A secondary structure analysis for the C-C’ loop during the course of the simulations are required to support the conclusion that terminal Gal residue triggers protein conformational changes.
- 8- Overall, the data related to MD simulations need to be deposited in public repositories (e.g., Zenodo) to allow readers and reviewers assess the quality and reproducibility of the simulations as mentioned in guidelines here: <https://pubs.acs.org/doi/10.1021/acs.jcim.3c00599>)

Minor points

- 1- The mention of programs in the figure legends is not necessary. This is more suited for the method section.
- 2- In Fig. 4g, it would improve readability if the labels “R122A” and “W127A” were placed directly near their respective spectra rather than requiring the reader to consult the legend (as done in Fig. 5b).
- 3- Legend figure 5: there appear to be two lines referring to panel b.

Version 1:

Reviewer comments:

Reviewer #1

(Remarks to the Author)

Dear Dr Fadda and co-workers,

You have addressed all my comments and have thus improved (in my opinion) an already excellent manuscript.

Reviewer #2

(Remarks to the Author)

The authors have fully addressed the issues raised in my previous review. I recommend that the manuscript be accepted for publication.

Reviewer #3

(Remarks to the Author)

The authors answers to all my comments. This manuscript is ready to be published.

We would like to thank all three reviewers for the time and effort they dedicated to examining our manuscript. We were very happy to read the positive comments and the constructive feedback. We took on board all the points raised and modified the manuscript accordingly (all changes are highlighted in the new draft). We hope that we were able to address all the comments exhaustively, both in the manuscript, and below in a detailed point-to-point response. In summary, we believe that the reviewers' feedback made our analysis of the results even more convincing and thus the manuscript stronger. For clarity, the Reviewers comments are in italics, separated into different section for each Reviewer. Our answers are preceded by '**Response:**' in bold, red fonts.

Response to Comments from Reviewer #1

Dear Dr. Fadda and co-workers,

I was very excited to read your manuscript on the surprising new binding mode of Siglec-6. I enjoyed the manuscript for its combination of computational structural biology, molecular dynamics simulations, and a state-of-the art set of actual binding experiments. As the 'siglec-mystery' keeps prevailing in the carbohydrate community, I think this is a very important, timely, and thorough addition to this field and therefore can only recommend its publication.

Response: Very much appreciated.

This is not without some comments. These are minor, as this paper has clearly been through review already. They mainly centre around one key question that is not mentioned in the discussion, and that is 'why has this binding mechanism evolved?'. What evolutionary reason could there be for evolving a binding mechanism that involves the membrane? I think including a few lines of text on what the author's take on this is, would be very nice.

Response: This is a very intriguing point of discussion we are happy to address in more detail. We think that the evolutionary reason is to enhance binding specificity for a specific type of ganglioside among many similar species, i.e. to turn a broadly specific, sialic acid-binding lectin into a highly selective GM1 binder. The results we obtained by native MS show that Siglec-6 cannot discriminate between different monosialylated oligosaccharide epitopes, binding GM1os with the same affinity as GM2os and GM3os. The relative

orientation of the Siglec-6 Ig V-set domain inserted in the bilayer through Trp127 and Lys126 (see new panels **Fig.2.h** and **2.k**), turns it into a GM1-specific binder, while it precludes binding to GM2os (and GM3os) due to loss of interaction with the -CC'-loop. In the discussion, we further speculated that the Siglec-6 membrane interactions could even be at the core of the Siglec-6 recognition mechanism, scanning the cell membrane surface to find its target epitope. An investigation into this area is beyond the scope of this work, possibly requiring a coarse-grained (CG) approach to MD simulations, ideally combined with super-resolution microscopy. We added the following sentences in the discussion section line 412-418 page 13:

“To this end, the evolutionary advantage of developing a mechanism that hinges on membrane interaction could rest in enhancing Siglec-6 selectivity for GM1, namely turning a sialic acid-binding lectin into a high-specificity molecular precision tool. The study of how such sophisticated scanning mechanism impacts epitope recognition and binding at the molecular level of detail would be very interesting, yet we believe to be beyond the scope of this work, and it may require the use of a coarse-grained MD approach, ideally in combination with super-resolution microscopy^{49,50}.”

The other minor point was that, for clarity, the authors include one or two lines more of text on their mass spec methods in the main text. I had to now look it up via the reference, which many readers may not want to do.

Response: Thank you for highlighting this. The acronym is now explained in the introduction (page 3 lines 118-119) and appropriately referenced. At page 9, line 299-303 we added the following brief description that we believe would be satisfactory to a broad readership:

“We used Concentration Independent (COIN)-Catch-and-Release (CaR)-nMS assay²⁶ to quantitatively screen free ganglioside oligosaccharides binders in unknown concentration conditions. This approach is based on the slow mixing of solutions inside a nanoESI emitter to achieve a nearly constant glycan concentration flux²⁶.”

Final point. Page 3, line 89 misses a space between the reference (15-18) and the word 'can'.

Response: The typo was corrected.

Response to Comments from Reviewer #2

Siglecs are immunoregulatory receptors that recognise sialylated glycoconjugates. In most of the 14 human Siglecs, binding of sialic acid involves a conserved arginine in the V-set domain. Here, using all-atom molecular dynamics (MD), mutagenesis, and binding assays with liposomes and nanodiscs, we show that Siglec-6 engages membrane-embedded GM1 through a mechanism that only partly depends on this canonical Arg122. MD reveals insertion of Trp127 into the lipid bilayer and contacts of Lys126 with phospholipid headgroups that supplement binding to the GM1 epitope; disrupting these residues markedly reduces binding to GM1 presented from membranes. In contrast, binding to free GM1/GM2/GM3 oligosaccharides is weakened by R122A and is insensitive to W127A, indicating Arg122-dependent recognition off-membrane. The data support a model in which Siglec-6 samples membranes via low-affinity lipid interactions and achieves specificity for GM1 when its terminal galactose engages the C–C' loop, allowing intermittent Arg122–Neu5Ac contacts. This work clarifies how Siglec-6 adapts sialoside recognition to the membrane environment. Overall, the manuscript is very well written and presents high-quality data. However, I have some (minor) points that should be clarified before publication:

Response: Very much appreciated.

- 1. The proposed mechanism depends on the V-set C–C' loop (and Lys126/Trp127) geometry. AF loop confidence (pLDDT) is often low for exposed loops. It would be helpful to show the per-residue AF confidence for the C–C' loop and the KW segment, perhaps in a supplementary figure.*

Response: This is a very good point. We added **Figure S.1** in the SI showing the structure of the Siglec-6 from AF with the standard colouring indicating the pLDDT thresholds and the per-residue confidence across the whole protein, highlighting the residues in the -CC'- loop (very high pLDDT>90) and KW segment (high pLDDT>70). To clarify this point, in the main manuscript we added the following sentence on page 5 lines 182 and 183:

“In this work we used the AlphaFold (AF)²⁹ model AF-O43699-F1 of Siglec-6 deposited in the EBI-EMBL AF protein structure database³⁰, with 69% of the residues predicted with very high confidence (pLDDT > 90) and 26% with high confidence (pLDDT > 70), see Fig. S.1.”

2. *Please provide a rationale for the chosen membrane composition used in the MD simulations, nanodisc, and liposome studies. It would also be useful to know whether Trp127 insertion and Arg122 intermittency persist with alternative membrane compositions, such as POPC:chol with or without sphingomyelin.*

Response: The composition of the membrane in the liposome and nanodisc studied was the result of extensive testing done in previous work (Schmidt et al. 2023). The composition of the membrane in the MD simulations was chosen to match this. This is now stated clearly in the manuscript, see page 3 lines 105 to 107:

“The membrane used in the simulations is set to match the experimental composition, selected based on the results of extensive testing in earlier work¹⁴ with varied lipid compositions and concentration of cholesterol.”

3. *The complex was seeded by aligning membrane GM1 to a sialosides analogue pose from Siglec-3 (PDB 7AW6), then aligning Siglec-6 to Siglec-3. Could the authors clarify whether the observed ligand orientation is driven by the membrane and KW contacts rather than inherited from the initial alignment?*

Response: This is an important point about epitope recognition in highly heterogeneous environments, which is the first questions we addressed at the beginning of the study, whether the presentation of the ligand and its orientation relative to the membrane was consistent with recognition of the sialic acid across all the monosialylated gangliosides series, independently of the presence of the Siglec-6. As described at the beginning of the Results section (page 5, lines 161-173) to investigate this point we ran independent MD simulations of membrane embedded GM1 and GM3. The analysis of the results show that the structure and orientation of the glycan epitopes is conserved throughout the trajectory, with a consistent presentation of the sialic acid epitope towards the solvent (see Fig. 2c and 2d). We built the complex around a representative structure of GM1 taken from this MD equilibrium ensemble. The structure of the Siglec-6 binding site complements this equilibrium conformation when the Ig V-set is

embedded in the membrane through KW contacts, supported by the interaction of the C-C' loop with the terminal Gal of GM1. When the contact between the KW and the membrane is lost the Siglec-6 detaches from the epitope. This detachment does not affect the epitope's structure. In summary, the ligand orientation and structure is determined by the epitopes' sequence and branching, not by Siglec-6 binding. To clarify this point we added the following sentence page 6 line 188-190:

“As an important note, the orientation of the Siglec-6 was selected to complement the chosen equilibrium conformation of the GM1 ligand embedded in the membrane and no alteration to that were made to enhance structure complementarity.”

4. *Fig. 2f/h qualitatively show intermittent contact defined with a 5 Å threshold. Please report occupancy (% time, mean lifetimes, and transition counts (± SEM across replicates) for the Arg122–Neu5Ac salt bridge. And also provide a free-energy proxy, e.g., potential of mean force (distance CV) or alchemical $\Delta\Delta G$ (R122A vs WT), to quantify how “dispensable” Arg122 is in membranes vs free oligosaccharides.*

Response: We added the results of a population analysis (i.e. 68.5% Arg122–Neu5Ac bound see page 6 lines 229-230) in the text and complemented the distance analysis in **Fig 2.f** and **2.g** with more analysis, namely : 1) Orientation of the Ig V-set domain in bound (GM1) and unbound (GM2) Siglec-6 through a tilt angle between the normal to the membrane plane and an auxiliary line indicating the orientation of the Ig V-set domain through its Ca principal component (PC)-1 eigenvector. 2) Calculation of the degree of insertion (%) into the membrane of different residues, showing the contributions to binding of the Trp127 and Lys 126 in both the non-glycosylated Siglec-6 and in the fully glycosylated Siglec-6. Evaluating free energies in this context is not possible because of the large error associated to FEP or TI methods (and the computational cost) of such approaches in highly flexible systems. Free energy end-point method, such as MM-GB(or PB)SA, are less computationally demanding but rather error prone in this context and require evaluation of the binding entropy separately. Ultimately, we did not see the need to perform such calculations as we have a complete set of experimental values produced for this and in previous work¹⁴. In our opinion, this is a typical case where free energy calculations are likely to be much more effort and time consuming, while inconclusive, than experiments.

5. *The authors simulate both non-glycosylated and fully glycosylated Siglec-6 but primarily report the former. Given the KW segment is near the V-set domain, N103 (the only V-set sequon) could influence loop dynamics. Please summarise whether the fully glycosylated model preserves KW membrane insertion and Arg intermittency. If possible, testing an N103Q mutant could clarify whether V-set glycosylation modulates GM1 binding.*

Response: The simulation of a N103Q mutant would not lead to any difference as it is way too subtle. But we ran two uncorrelated replicas of the fully glycosylated Siglec-6 in complex with GM1 and some of the results are now included in Fig.2 for comparison. In summary, glycosylation does not alter the binding mechanism, but we find that the N103 glycan in particular with N149 stabilize the interaction between the Ig V-set and the adjacent C2-set domain, limiting their relative mobility, likely straightening the Ig V-set approach onto the membrane, as shown by the tilt angle analysis in Fig.2.h. If this may result in a change of binding affinity, it would be an interesting question, which may require a targeted experiment with additional glycoproteomics/glycomics analysis, which we believe may be beyond the scope of this work.

6. *Minor typo (p. 10): “GM1-3os ... GM2os and GM2os” should read GM1os, GM2os, GM3os.*

Response: Corrected.

Response to Comments from Reviewer #3

(.) Overall, this is an interesting manuscript that combine Molecular Dynamics (MD) simulations and biochemical assays to shed new lights on lectin-glycolipid interactions. Specifically, S. D’Andrea et al. identified new residues (K126 and W127) at the surface of Siglec-6 involved in direct interactions with phospholipids. The biochemical assays coupled to mutagenesis appear to be in good agreement with theoretical results.

Response: Very much appreciated.

However, the MD analyses would benefit from a more extensive and quantitative treatment of dynamic processes observed throughout the simulations, rather than focusing on the final frames. This would significantly strengthen the theoretical part of the study. In addition, placing the work in the broader context of previously published MD studies (both atomistic and coarse-grained) in the Introduction and Discussion would help readers appreciate what can currently be achieved using MD simulations.

Major points:

- 1. In introduction, a paragraph summarizing theoretical works (and especially MD simulations) on protein-sugar or protein-lipid interactions would help the readers to understand what it is possible to do with this type of approach.*

Response: We absolutely agree with the point raised here and added the following paragraph to the Introduction (page 3 pages 95-105), hopefully filling the gap of information, as the reviewer suggested, while making the introduction more comprehensive and informative to a broader readership.

“Molecular dynamics (MD) simulations have been used extensively and successfully to investigate at the atomistic and molecular levels of details the distribution and dynamics of gangliosides, and other glycolipids, in lipid bilayers¹⁹⁻²¹. Coarse graining (CG) methods have been particularly useful to explore the enormous complexity of such highly dynamic systems at biologically relevant timescales, where membranes can bear a wide range of chemically diverse lipids, in different concentrations, with different structure flexibilities and diffusion properties²². Although the CG force fields sophistication has been greatly improved recently²³, allowing users to obtain important insights into membrane composition, structure and biology^{24,25}, in this work we needed an atomistic approach for our simulation to characterise the Siglec-6 recognition mechanism and to understand how this facilitates a discrimination between structurally similar monosialylated gangliosides.”

- 2. The authors should clarify the source of the lipid composition used for the membrane (60% DSPC + 40% cholesterol) and how it relates to biological membranes. From my experience, cellular membranes are often asymmetric (outer and inner leaflets differ in composition), which is not the case here. Why was asymmetry not considered*

? With such a high cholesterol concentration, the membrane may be in a gel phase (see Fig. 2a and 2g). Is this a behaviour the authors intended to reproduce? How might this affect GM1 sugar moiety exposure and Siglec-6 interactions with the membrane? All these points need, at least, to be discussed.

Response: The composition of the membrane in the liposome and nanodisc studied was the result of extensive testing done in previous work (Schmidt et al. 2023). The composition of the membrane in the MD simulations was chosen to match this. This is now stated clearly in the manuscript, see page 3 lines 105 to 107:

“The membrane used in the simulations is set to match the experimental composition, selected based on the results of extensive testing in earlier work¹⁴ with varied lipid compositions and concentration of cholesterol.”

3. *How the construction of the system can influence the final results? In particular, did the authors consider using coarse-grained MD simulations to study the full process of interaction of siglec-6 with the membrane by placing the protein away from the membrane and allowing it to diffuse. All the models to perform such simulations are readily available, and computationally less expensive than atomistic MD simulations. This should, at least, be discussed.*

Response: This is an important point that we were very careful to consider in the simulation set up, making sure the presentation of the ligand and its orientation relative to the membrane was consistent with recognition of the sialic acid across all the monosialylated gangliosides series, independently of the presence of the Siglec-6. As described at the beginning of the Results section (page 5, lines 161-173) to investigate this point we ran independent MD simulations of membrane embedded GM1 and GM3. The analysis of the results show that the structure and orientation of the glycan epitopes is conserved throughout the trajectory, with a consistent presentation of the sialic acid epitope towards the solvent (see Fig. 2c and 2d). We built the complex around a representative structure of GM1 taken from this MD equilibrium ensemble. The structure of the Siglec-6 binding site complements this equilibrium conformation when the Ig V-set is embedded in the membrane through KW contacts, supported by the interaction of the C-C' loop with the terminal Gal of GM1. When the contact between the KW and the membrane is lost the Siglec-6 detaches from the epitope.

This detachment does not affect the epitope's structure. In summary, the ligand orientation and structure is determined by the epitopes' sequence and branching, not by Siglec-6 binding. To clarify this point we added the following sentence page 6 line 188-190:

“As an important note, the orientation of the Siglec-6 was selected to complement the chosen equilibrium conformation of the GM1 ligand embedded in the membrane and no alteration to that were made to enhance structure complementarity.”

Furthermore, regarding CG simulations, the idea of an MD simulation connecting recognition and binding in this case is great; yet in the absence of an experimental back-up, e.g. super-resolution microscopy, it would be difficult to verify. We added the following sentences in the discussion section page 13, line 412-418:

“To this end, the evolutionary advantage of developing a mechanism that hinges on membrane interaction could rest in enhancing Siglec-6 selectivity for GM1, namely turning a sialic acid-binding lectin into a high-specificity molecular precision tool. The study of how such sophisticated scanning mechanism impacts epitope recognition and binding at the molecular level of detail would be very interesting, yet we believe to be beyond the scope of this work, and it may require the use of a coarse-grained MD approach, ideally in combination with super-resolution microscopy^{49,50}.”

Moreover, based on the availability of high-resolution X-ray diffraction structures clearly showing the conserved Arg binding mode and the high degree of sequence similarity across CD33-related Siglecs, it seemed to us more of a speculation to build a complex based on a CG simulation of a protein-ligand encounter rather than on crystallographic evidence. Based on the results of all our simulations, we are quite confident that the building of the complex did not affect the results, which ultimately are clearly supported by all experimental evidence.

4. *The distance analysis presented in Fig. 2-f is quite a simple proxy for identifying siglec-6 interactions with the membrane. Can a slightly more sophisticated and exhaustive analysis provide additional details on which surface residues, beyond the ones described by the authors, are involved? Tools such as MDAnalysis can perform facilitate such analyses. Furthermore, this type of analysis should also to be performed for siglec-6 in interaction in GM1, as presented in fig. 2-a.*

Response: As suggested we calculated the % of insertion of any residue in the Siglec-6 (glycosylated and not) through a plane passing through the phosphate groups (selected as $z=0$) and measuring the relative position (depth) of the COM of all sidechains in proximity. The results are shown in Fig. 2.k and show the dominant contributions of both Tyr126 and Lys126. The only residue that we did not mention before is Tyr61, which COM transitions only sporadically (and only across the surface) when the Siglec-6 is membrane bound, contributing much more when the Siglec-6 detaches from the GM1 (at the end of both MD simulations). The time evolution used to calculate the bar plots is shown in Fig.S.2 and shows clearly this point. We did notice this intermittent contact of Tyr61 just by looking at the trajectories, but it did not appear to us to be sufficiently stable to contribute significantly to the binding.

5. *Analysing siglec-6 tilting (has done for GM1) in interaction with GM1 and GM2 can help assess the flexibility, the dynamic, and the stability of the system.*

Response: This is a great suggestion. We adapted the tilt angle analysis and the results are shown in Fig.2.h for the non-glycosylated Siglec-6 in complex with both GM1, and GM2 and for the glycosylated Siglec-6 in complex with GM1.

6. *A proper analysis of hydrogen bonds and salt bridge stability over time is necessary to quantify the interactions of siglec-6 with sugar moieties and lipids. MDAnalysis or VMD can easily perform such analysis.*

Response: The analysis is in Fig.2.f and 2.g (renumbered now). We added in the text that the salt bridge is formed 68.5% of the time along the trajectory.

7. *A secondary structure analysis for the C-C' loop during the course of the simulations are required to support the conclusion that terminal Gal residue triggers protein conformational changes.*

Response: The secondary structure analysis is shown in **Fig.S.2**. The results demonstrate that the loop is disordered, while a slight conformational transition (coil to turn) is indeed linked to the binding of the C-C' loop to the Gal. We apologize if this was not clear, but we never

meant to imply that the terminal Gal triggers a conformation change of any kind, but rather that the C-C' loop regulates binding specificity, as it is known to do across Siglecs.

8. *Overall, the data related to MD simulations need to be deposited in public repositories (e.g., Zenodo) to allow readers and reviewers assess the quality and reproducibility of the simulations as mentioned in guidelines here: <https://pubs.acs.org/doi/10.1021/acs.jcim.3c00599>)*

Response: At the time of the first submission all trajectories were shared OA on our repository on GlycoShape (<https://glycoshape.org/downloads>), but somehow that note went missing. We added copies of the simulations on Zenodo (<https://doi.org/10.5281/zenodo.17312619>) according to the suggestion.

Minor points:

9. *The mention of programs in the figure legends is not necessary. This is more suited for the method section.*

Response: We completely agree the caption may sound very detailed and somewhat redundant, but some journals do require such level of detail. We will keep the comment in mind, while waiting until proofs to see what the journal's formatting standards are, i.e. if they require us to remove that information, we'll be happy to do that.

10. *In Fig. 4g, it would improve readability if the labels "R122A" and "W127A" were placed directly near their respective spectra rather than requiring the reader to consult the legend (as done in Fig. 5b).*

Response: Done.

11. *Legend figure 5: there appear to be two lines referring to panel b*

Response: Apologies, typo corrected.

Schmidt, Edward N., Dimitra Lamprinaki, Kelli A. McCord, Maju Joe, Mirat Sojitra, Ayk Waldow, Jasmine Nguyen, et al. 2023. "Siglec-6 Mediates the Uptake of Extracellular Vesicles through a Noncanonical Glycolipid Binding Pocket." *Nature Communications* 14 (1): 2327.